# Risk factors and survival in patients with COVID-19 in northeastern Brazil

Ana Tereza Fernandes[ID][1][¤]*, Eujessika K. Rodrigues[2], Eder R. Araújo[1], Magno F. Formiga[3‡], Priscilla K. Sá Horan[4‡], Ana Beatriz Nunes de Sousa Ferreira[1], Humberto A. Barbosa[5‡], Paulo S. Barbosa[2‡]

1 Department of Physical Therapy, State University of Paraiba, Campina Grande, Brazil, 2 Center of Technology Strategies in Health, State University of Paraiba, Campina Grande, Brazil, 3 Department of Physical Therapy, Federal University of Ceara, Fortaleza, Brazil, 4 Don Luiz Gonzaga Fernandes Trauma and Emergency Hospital, Campina Grande, Brazil, 5 Federal University of Alagoas, Maceio, Brazil

☯ These authors contributed equally to this work.
¤ Current address: Physical Therapy Department, Campina Grande (PB), Brazil
‡ These authors also contributed equally to this work.
* aninhat.sales@gmail.com

**Data Availability Statement:** The information about data availability is at link https://figshare.com/s/439d0d45b7b3e442728f.

## Abstract

### Background

Knowledge about the epidemiology and risk factors surrounding COVID-19 contributes to developing better health strategies to combat the disease.

### Objective

This study aimed to establish a survival analysis and identify the risk factors for patients with COVID-19 in an upper middle-income city in Brazil.

### Methods

A retrospective cohort study was conducted with 280 hospitalized patients with COVID-19. The eCOVID platform provided data to monitor COVID-19 cases and help the communication between professionals.

### Results

Age $\geq$ 65 years was associated with decreased survival (54.8%), and females had a lower survival rate than males (p = 0.01). Regarding risk factors, urea concentration (p<0.001), hospital length of stay (p = 0.002), oxygen concentration (p = 0.005), and age (p = 0.02) were associated with death.

### Conclusion

Age, hospital length of stay, high blood urea concentration, and low oxygen concentration were associated with death by COVID-19 in the studied population. These findings corroborate with studies conducted in research centers worldwide.

**Funding:** Yes.This study was financed by the Coordenação de Aperfeiçoamento de Pessoal de Nível Superior – Brasil (CAPES) (Finance Code CAPES EPIDEMIAS 09/2020-grand/award number:23038.013745/2020-69 to H.A.B) and Fundação de Apoio à Pesquisa do Estado da Paraíba – Brasil (FAPESQ) (Finance Code COVID-19 003/2020). The funders The funders had role in study design, decision to publish and preparation of the manuscript.

**Competing interests:** The authors have declared that no competing interests exist.

## Introduction

A transmissible disease caused by the novel coronavirus (SARS-CoV-2) has increased the number of severe acute respiratory syndrome cases worldwide since December 2019. Although efforts were made to determine the prevalence and factors associated with the epidemiology and severity of COVID-19 [1,2], more than 551 million people were infected worldwide until July 2022, and more than four million died. In the Americas, more than 164 million cases have been confirmed, whereas Brazil has accumulated more than 32 million cases [3].

Many countries have focused on defining and identifying strategies to fight the disease, prevent hospitalizations, and maintain economic activities [4–6]. One critical point of the disease is its form of presentation, which varies from asymptomatic to very mild or critical symptoms. Symptoms may also persist even after the acute phase, and individuals who initially did not report or have mild symptoms may evolve to sudden health decline or death [4,7,8].

In this sense, identifying factors predisposing to a high risk of hospitalization and death contribute to preventive measures, especially in developing countries facing difficulties in establishing an early diagnosis [8]. Therefore, this study describes the clinical and demographic characteristics, comorbidities, outcomes, survival, and factors associated with mortality of hospitalized patients with COVID-19 in an upper-middle-income city in Brazil.

## Methods

### Study design and participants

A retrospective cohort study was conducted with patients hospitalized due to COVID-19. Data were collected from the virtual platform eCOVID [2] between January-May 2021, developed by the Center for Strategic Health Technologies of the State University of Paraíba (NUTES/ UEPB) in Brazil. Qualified professionals obtained data during the admission of patients in two hospitals in the city. The study was approved by the research ethics committee of the Universidade Estadual da Paraíba (approval number 4.241.971) and followed all ethical aspects involving research in humans and the Declaration of Helsinki. The informed consent form was waived by the ethics committee.

We included hospitalized patients of both sexes, aged 18 or over, diagnosed with COVID-19 using RT-PCR test (nasopharynx swab) [3], and with epidemiological history of COVID-19 and/or relevant clinical symptoms and serological parameters [9,10]. Patients whose COVID-19 diagnosis was discarded or outcomes were not recorded were excluded.

### eCOVID platform

The eCOVID platform (ecovid.nutes.uepb.edu.br) was developed to store patient data (e.g., symptoms, comorbidities, vital signs, and laboratory and imaging exams) and facilitate communication among professionals treating patients with COVID-19. The platform allows remote assistance/communication among professionals working in the public Brazilian Unified Health System and the private sector. It also provides a database for epidemiological studies in the city where the study was conducted.

### Variables and data collection

The following variables were included for data analysis: (1) profile of patients (age and sex); (2) diagnostic method (RT-PCR, immunoglobulin serology, or clinical-epidemiological); (3) symptoms presented on admission; (4) comorbidities; (5) vital signs; (6) laboratory and imaging exams; (7) risk stratification (subjectively collected by the medical doctor on admission); (8) hospital length of stay (hospital LOS), representing the total number of days of hospital

stay; (9) length of invasive mechanical ventilation (IMV); and (10) clinical outcomes (discharge or death). We also obtained laboratory parameters from medical records, such as blood cell count and renal function (i.e., urea and creatinine concentration).

Data from the first (admission) to the last day of hospitalization (hospital discharge or death) were included for analysis. Vital signs, symptoms, and laboratory tests were considered if obtained on admission, while data regarding IMV, hospital LOS, and imaging tests were obtained when the outcome (discharge or death) was recorded.

## Data analysis

All statistical analyses were conducted using the SPSS software, version 22 (SPSS, Inc., Chicago, IL, USA).

Categorical data were presented as relative frequency and numerical data as mean and standard error (SE). Patients were divided into survivors (patients discharged) and non-survivors (patients who died). Kolmogorov-Smirnov test assessed data normality, and the unpaired t-test compared data between groups. The chi-square test ($Chi^2$) compared comorbidities, signs, and symptoms between groups. The Cox (proportional hazards) regression investigated the effects of age ($< 65$ and $\geq 65$ years) and sex on death, whereas the survival analysis was performed using the Kaplan-Meier method. We also computed stepwise logistic regression models using the progressive selection technique (i.e., forward selection), in which only significant variables ($p \leq 0.05$) and the predictive value remained in the model to identify those interfering with death. A significance level of $p \leq 0.05$ was used for all analyses. Since the CDC stated that adults aged $\geq 65$ years were at greater risk of COVID-19 complications and death (https://www.cdc.gov/aging/covid19-guidance.html), we have decided to use this cut-off point to stratify individuals into younger vs. older adults.

Missing data were managed using multiple imputations by replacing missing cells with random values from a statistical model based on data distribution and underlying assumptions regarding the nature of the missing data. Five imputed datasets were created, and the final estimates considered the average of different sets of imputed estimates and SE. The multiple imputation approach was chosen because it generates more accurate values than those generated by single imputation methods.

## Results

A total of 365 patients were registered on the eCOVID platform; however, only the records of patients who had a positive RT-PCR test for COVID-19 (N = 280) were included in the study.

### Clinical characteristics

Table 1 shows the general and clinical characteristics, symptoms, and comorbidities presented by the studied population. Among 280 hospitalized patients with COVID-19 included in our study, 192 were discharged, and 88 died. Approximately 54% of patients were males with mean age of 62.42 years (SE: 1.45). The most common comorbidities were systemic arterial hypertension (SAH) (53%), diabetes mellitus (67%), and obesity (24%). Moreover, 10% of patients had other types of heart disease. Dyspnea was the most reported symptom (82%), followed by cough (59%), fever (57%), muscular pain (22%), headache (15%), and anosmia (11%). Furthermore, 27% of patients were critically ill, and 57% were moderately ill on admission.

Age (p = 0.001), dyspnea (p = 0.003), and presence of other heart diseases (p = 0.005) were different between survivors and non-survivors. SAH was the most reported comorbidity in the

**Table 1. Clinical characteristics, laboratory tests, signs and symptoms, and comorbidities.**

| | *Total (N = 280)* | *Survivors (N = 192)* | *Non-survivors (N = 88)* | *p-value* |
|---|---|---|---|---|
| **Clinical characteristics** | | | | |
| Sex (M/F) % | 54/46 | 55/45 | 53/47 | - |
| Age (years) | 62.42 (1.45) | 58.54 (1.22) | 70.87 (1.64) | <0.001* |
| RR (rpm) | 23.07 (0.52) | 22.51 (0.70) | 24.29 (0.93) | 0.150 |
| HR (bpm) | 86.48 (1.08) | 87.82 (1.25) | 83.56 (2.21) | 0.094 |
| SBP (mmHg) | 132.7 (1.61) | 134.78 (1.97) | 128.18 (3.03) | 0.068 |
| SpO$_2$ (%) | 93 (0.37) | 95 (0.27) | 91 (0.94) | <0.001* |
| Laboratory tests | | | | |
| Hemoglobin (g/dL) | 12.68 (0.12) | 12.83 (0.14) | 12.36 (0.23) | 0.077 |
| Leukocytes (mm$^3$) | 11.47 (0.36) | 10.30 (0.40) | 14.03 (0.69) | <0.001* |
| Platelets (mm$^3$) | 269.4 (6.86) | 276 (8.57) | 255 (11.87) | 0.168 |
| Creatinine (mg/dL) | 1.46 (0.14) | 1.32 (0.17) | 1.76 (0.21) | 0.127 |
| Urea (mg/dL) | 58.32 (46.47) | 44.52 (2.43) | 85.28 (6.51) | <0.001* |
| Signs and Symptoms[†] | | | | |
| Days to hospitalization (days) | 8.40 (0.30) | 8.79 (0.37) | 7.55 (0.50) | 0.057* |
| Dyspnea (%) | 82 | 77 | 92 | 0.003 |
| Cough (%) | 59 | 60 | 58 | 0.925 |
| Fever (%) | 57 | 58 | 56 | 0.838 |
| Muscular pain (%) | 22 | 25 | 16 | 0.102 |
| Headache (%) | 15 | 15 | 14 | 0.800 |
| Ageusia (%) | 6 | 7 | 6 | 0.934 |
| Anosmia (%) | 11 | 15 | 5 | 0.019* |
| Runny nose (%) | 2 | 3 | 2 | 0.991 |
| Sore throat (%) | 2 | 3 | 3 | 0.991 |
| Asthenia (%) | 10 | 12 | 6 | 0.126 |
| Joint pain (%) | 1 | 1 | 1 | 0.579 |
| Nausea and/or vomit (%) | 6 | 8 | 5 | 0.543 |
| Fatigue (%) | 2 | 4 | 1 | 0.433 |
| Comorbidities[††] | | | | |
| Number of comorbidities | 1.40 (0.67) | 1.28 (0.07) | 1.66 (0.12) | 0.007* |
| SAH (%) | 53 | 55 | 53 | 0.787 |
| Diabetes (%) | 67 | 30 | 39 | 0.178 |
| Obesity (%) | 24 | 23 | 27 | 0.522 |
| COPD (%) | 3 | 3 | 5 | 0.489 |
| Others cardiopaties (%) | 10 | 6 | 17 | 0.005 |
| CKD (%) | 4 | 4 | 5 | 0.977 |
| Athsma (%) | 2 | 3 | 2 | 0.804 |
| **Clinical Evolution** | | | | |
| Hospital LOS (days) | 6.02 (0.40) | 5 (0.95) | 8.81 (0.92) | <0.001* |
| IMV duration (days) | 1.74 (0.38) | 0.53 (0.33) | 4.38 (0.65) | <0.001* |

M: Male, F: Female, RR: Respiratory rate, HR: Heart rate, SBP: Systolic blood pressure, SpO$_2$: Peripheral oxygen saturation, SAH: Systemic arterial hypertension, CKD: Chronic kidney disease, IMV: Invasive mechanical ventilation, LOS: Length of stay, COPD: Chronic obstructive pulmonary disease, rpm: Respiration per minute, bpm: Beats per minute, mmHg: Millimeters of mercury, %: Percentage, g: Grams, mg: Milligrams, dL: Deciliters, mm: Millimeters. Data presented as mean and standard error.

*Comparison between groups using Chi$^2$ test (categorical variables) or unpaired t-test (numerical variables).

[†]Proportions represent the number of people who presented that signs or symptoms.

[††]Proportions represent the number of people who had that comorbidity.

non-survivor group (53%); this group also presented a high prevalence of cough (58%) and fever (56%).

We observed significant differences in $SpO_2$ (p < 0.001), leukocyte count (p < 0.001), blood urea concentration (p < 0.001), anosmia (p = 0.019), number of associated comorbidities (p = 0.007), hospital LOS (p < 0.001), length of IMV (p < 0.001), and time between symptom onset and hospitalization (p = 0.057).

### Survival analysis

The overall model, including age and sex as predictors, significantly improved the fit compared with the null model [$\chi2$ (2) = 8.395, p = 0.01]. With increasing age, the hazard associated with death tended to be greater, even though this predictor was not significant ($\beta$ = 0.424, p = 0.07). On the other hand, sex was an independent and significant predictor of mortality hazard ($\beta$ = 0.461, p = 0.03), with time until death higher for males than females.

The survival analysis showed that 81.4% of patients aged < 65 years survived (Fig 1A). The survival rate of patients aged $\geq$ 65 years reduced to 54.8% and reached 50% in eleven days of hospitalization. In contrast, patients aged < 65 years reached a survival rate of 50% only on the 14th day of hospitalization (p = 0.04). Regarding sex, 67.7% of females and 69.3% of males survived. Females and males reached a 50% survival rate on the 12th and 17th days of hospitalization (p = 0.01), respectively (Fig 1B).

### Risk factors associated with mortality

The stepwise logistic regression with a forward selection model included all numerical variables assessed (Table 2). When grouped, urea concentration (p < 0.001), hospital LOS (p = 0.002), $SpO_2$ (p = 0.005), and age (p = 0.02) were better associated with death (Hosmer and Lemeshow test, p = 0.50). These variables were also different between groups. Fig 2 shows the ROC curve for the logistic regression model (AUC: 0.847, SE: 0.26, p < 0.00).

## Discussion

This study analyzed data from 280 hospitalized patients with COVID-19. The most common comorbidities were SAH, diabetes mellitus, and obesity, while most reported symptoms were dyspnea, cough, and fever. Age $\geq$ 65 years, low $SpO_2$, high urea concentration, and prolonged hospital LOS were associated with increased risk of death in these patients. Moreover, females had a low survival rate when hospital LOS lasted more than eleven days.

Hospitalization due to COVID-19 is linked to worsening symptoms and increased risk of developing severe acute respiratory syndrome, which increased in Brazil between 2019 and 2020 [11]. Thus, identifying the clinical characteristics of patients who reach severe clinical conditions is a global concern.

Casas-Rojo et al. [12] analyzed data from 15,111 hospitalized patients in 150 hospitals in Spain and found a high prevalence of males in the sample (more than 57% aged > 60 years). These authors also observed that SAH (50.9%), obesity (21.2%), and diabetes mellitus (19.4%) were the most prevalent comorbidities, while the most reported signs and symptoms on admission were fever (63.4%), dry cough (58%), and dyspnea (57.6%). The results of our study were similar to those observed in the Spanish population [12], suggesting a pattern of clinical characteristics of hospitalized patients with severe COVID-19. Similarly, Cabrini et al. [13] evaluated data from 1,591 patients admitted to intensive care units in Italy and found that most hospitalized patients were males (82%) with SAH (49%). Furthermore, 89% of patients aged > 64 years needed IMV, increasing the mortality rate and hospital LOS [11]. In our study, 28% of patients received IMV, and those who died needed this type of support for a

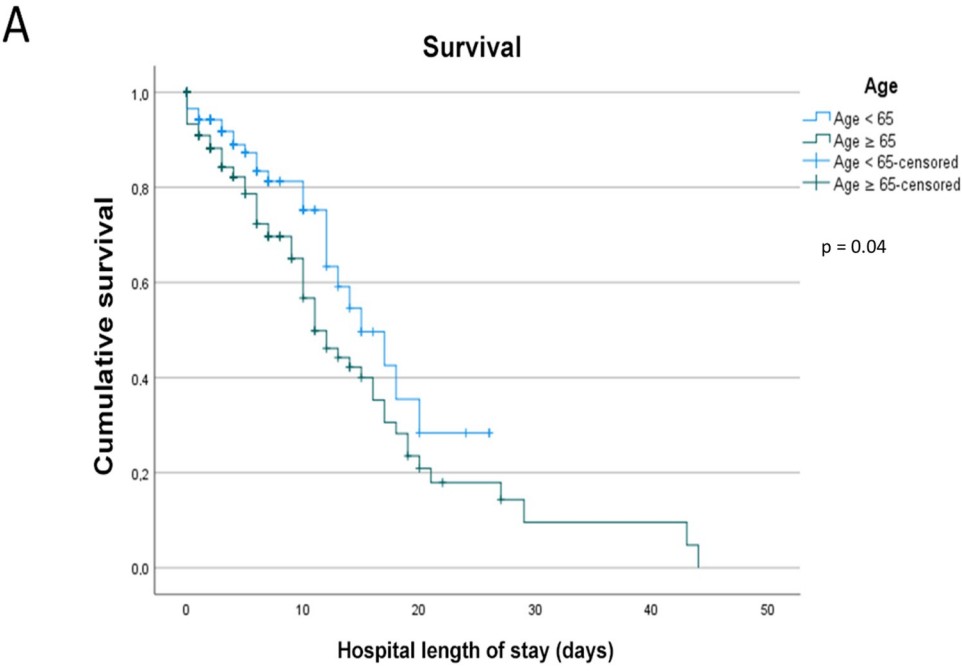

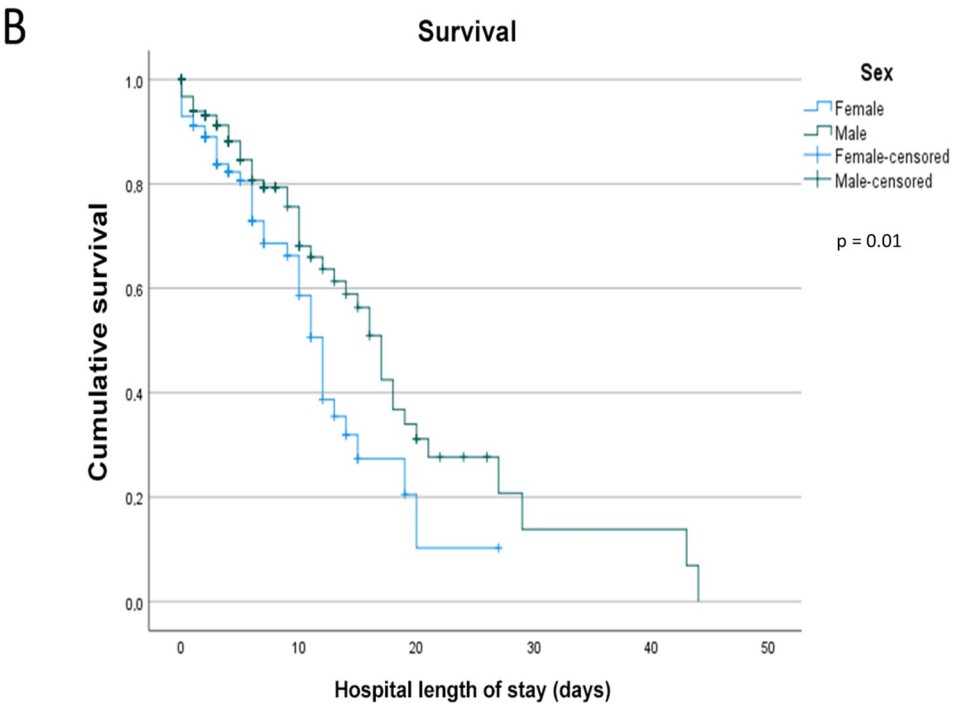

**Fig 1.** Survival analysis by (a) age and hospital length of stay and (b) sex and hospital length of stay.

**Table 2. Logistic regression of variables associated with death.**

| | RISK OF DEATH | | | |
|---|---|---|---|---|
| | β | p-value | OR | 95%CI |
| Age | 0.02 | 0.02 | 1.02 | 1.00–1.05 |
| SpO$_2$ | -0.11 | 0.005 | 0.89 | 0.82–0.96 |
| Hospital LOS | 0.10 | 0.002 | 1.10 | 1.03–1.18 |
| Urea | 0.03 | <0.001 | 1.03 | 1.02–1.04 |
| Constant | 5.18 | 0.15 | 177.94 | |

OR: Odds ratio, 95%CI: Confidence interval, SpO$_2$: Peripheral oxygen saturation, LOS: Length of stay.

longer period (4.38 days). We highlight that prolonged hospital LOS was associated with a high risk of death.

The role of comorbidities in the worsening of COVID-19 has been discussed in some studies. A systematic review by Bradley et al. [14] showed associations between diabetes and COVID-19 severity, progression to acute respiratory distress syndrome, in-hospital mortality, and need for IMV. Persistent hyperglycemia caused by diabetes may worsen the inflammatory

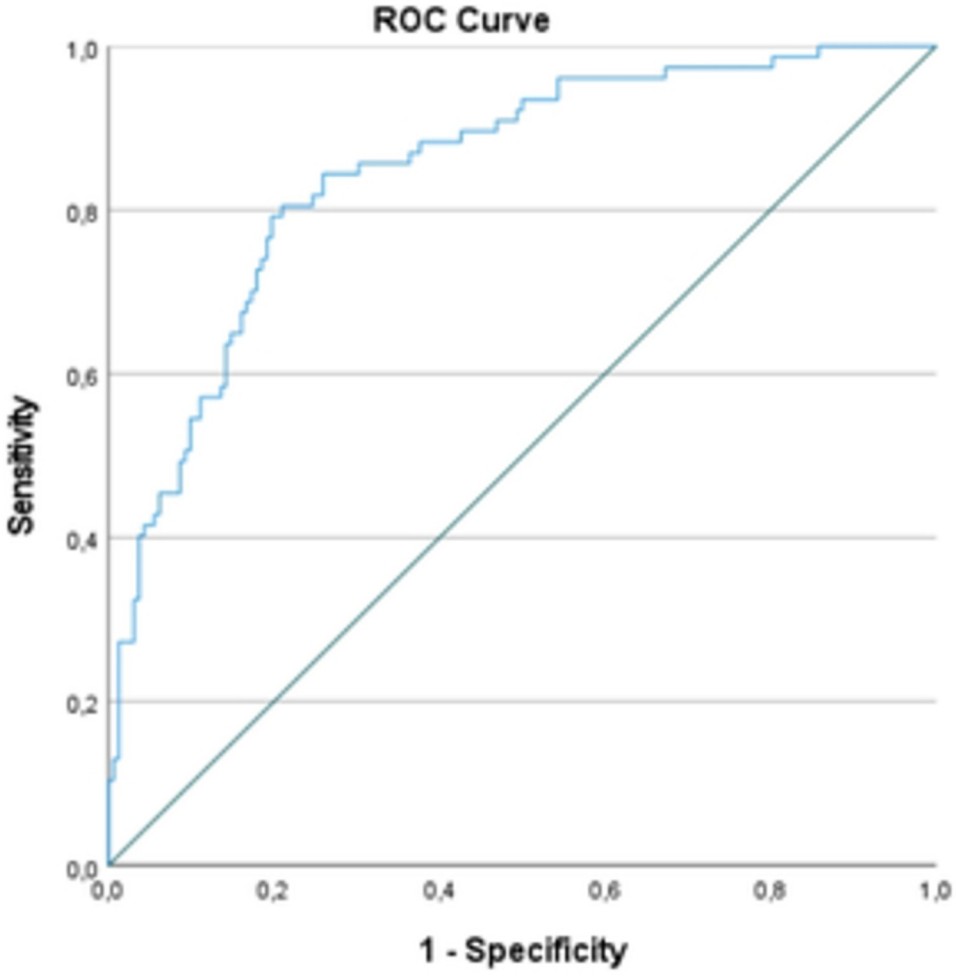

**Fig 2. ROC curve for the logistic regression model.**

condition and immune system through oxidative stress. Other diseases were also prevalent in patients hospitalized due to COVID-19, such as SAH (prevalence of 16.37%) and other cardiovascular diseases (prevalence of 12.11%). As the immune system does not recognize the protection pathways for this new virus, individuals with pre-existing diseases may become more vulnerable to the severe forms of COVID-19 [15].

Although studies showed a high prevalence of SARS-CoV-2 or other types of coronavirus in males [16,17], the survival rate dropped faster in females than males and remained low until the end of the study. The survival analysis conducted by Salinas-Escudero et al. [17,18] with 133 Mexican patients showed a higher mortality in older females, especially those aged > 75 years. We also observed a higher mortality rate in females than males (32.3% vs. 30.7%) but without significant a difference between groups (p = 0.77), probably due to the high rate of deaths unrelated to sex particularities.

Studies showing age as a risk factor for worsening COVID-19 are frequent in the literature [12,13,19]. Li et al. [20] showed that age ≥ 65 years was associated with risk of severe disease and SAH. In our study, the survival rate in females aged ≥ 65 years declined to 50% when hospitalization was longer than eleven days. Also, the mean age of patients who died was 70.87 years, 12 years more than those discharged. Although age is still a controversial risk factor for COVID-19, its role in mortality is well established [21]. In this sense, are the aspects inherent to aging responsible for developing the disease, or is the presence of individual factors (e.g., immune response), comorbidities, and particular aspects of older adults that worsen the response to the viral infection? As already established, chronic diseases are common in older adults and may cause health [16] problems, especially during the COVID-19 infection.

Other factors, such as $SpO_2$, hospital LOS, and blood urea concentration, were associated with mortality in our sample. Blood urea concentration was also associated with COVID-19 severity in the study by Ok et al. [22]. These authors observed that patients who progressed to severe disease had higher urea concentrations than those with moderate disease. Moreover, urea/creatinine ratio, white cell count, C-reactive protein, monocyte/lymphocyte ratio, and neutrophil/lymphocyte ratio were considered predictive factors for disease severity. These findings reinforce that early assessments may identify patients at risk of disease worsening.

Prolonged hospitalization, mainly in older adults, was associated with mortality. Thai et al. [19] showed that patients who died had a mean hospital LOS of eight days, whereas those who survived had a hospital LOS of six days. Prolonged hospitalization may impair muscle strength, quality of life, and functional capacity of older adults [23]. Also, the negative impacts of COVID-19 on physical, cognitive, and social well-being of patients who recovered from prolonged intensive care unit or hospital LOS due to acute respiratory illnesses are already recognized [24].

The limited access of the population to the COVID-19 diagnostic test and its acquisition and availability by the health system may be a potential limitation of this study. Specifically, during hospital stay, some patients were already outside the window of time to detect SARS-CoV-2 using RT-PCR. Therefore, the diagnosis was performed by identifying the immunoglobulins M and G and using clinical and/or radiological evolutions of the patient. The acceptance and use of the eCOVID platform can be considered another limitation of the study. We believe some professionals did not adhere to the platform, which may have caused losses in the registration of patients within hospital units that use the platform. Furthermore, the lack of information about signs and symptoms of COVID-19 may have delayed health care seeking. As a result, patients arrived at the hospital presenting advanced forms of the disease. In this sense, the Unified Health System should accelerate the care for those most vulnerable to developing the severe forms of the disease by mapping and referring symptomatic patients, older adults, and individuals with pre-existing comorbidities to care facilities; thus, avoiding hospitalizations and worse prognoses.

## Conclusions

Age, hospital LOS, high blood urea concentration, and low $SpO_2$ were associated with mortality by COVID-19 in the evaluated population. Considering that several variables associated with increased mortality are assessed at hospital admission, this study may guide the development of public health interventions to prevent the clinical evolution of COVID-19 to severe conditions. Furthermore, health professionals can assist and contribute to prevention by early identifying patients with severe COVID-19 in primary care.

## Acknowledgments

The authors thank Probatus Academic Services for providing scientific language revision and editing.

## Author Contributions

**Conceptualization:** Ana Tereza Fernandes, Eujessika K. Rodrigues, Priscilla K. Sá Horan, Humberto A. Barbosa, Paulo S. Barbosa.

**Formal analysis:** Ana Tereza Fernandes, Eujessika K. Rodrigues, Magno F. Formiga, Humberto A. Barbosa, Paulo S. Barbosa.

**Funding acquisition:** Humberto A. Barbosa, Paulo S. Barbosa.

**Investigation:** Eder R. Araújo, Priscilla K. Sá Horan, Ana Beatriz Nunes de Sousa Ferreira.

**Methodology:** Ana Tereza Fernandes, Eujessika K. Rodrigues, Eder R. Araújo, Magno F. Formiga, Priscilla K. Sá Horan, Ana Beatriz Nunes de Sousa Ferreira, Humberto A. Barbosa, Paulo S. Barbosa.

**Project administration:** Ana Tereza Fernandes, Humberto A. Barbosa, Paulo S. Barbosa.

**Writing – original draft:** Ana Tereza Fernandes, Magno F. Formiga, Paulo S. Barbosa.

**Writing – review & editing:** Paulo S. Barbosa.

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
