## [Decision Letter · Decision Letter 0]

22 Jun 2022

PONE-D-22-12861Risk factors and survival in patients with COVID-19 in northeastern Brazil.PLOS ONE

Dear Dr. Fernandes,

Thank you for submitting your manuscript to PLOS ONE. After careful consideration, we feel that it has merit but does not fully meet PLOS ONE’s publication criteria as it currently stands. Therefore, we invite you to submit a revised version of the manuscript that addresses the points raised during the review process.

Please pay attention to the comments raised about your paper and try your best to address every one of them==============================

We look forward to receiving your revised manuscript.

Kind regards,

Claudio Andaloro

Academic Editor

PLOS ONE

Journal Requirements:

[NO authors have competing interests]. 

Reviewers' comments:

Reviewer's Responses to Questions

**Comments to the Author**

1. Is the manuscript technically sound, and do the data support the conclusions?

Reviewer #1: Yes

Reviewer #2: Partly

2. Has the statistical analysis been performed appropriately and rigorously? 

Reviewer #1: No

Reviewer #2: Yes

3. Have the authors made all data underlying the findings in their manuscript fully available?

Reviewer #1: No

Reviewer #2: Yes

4. Is the manuscript presented in an intelligible fashion and written in standard English?

Reviewer #1: Yes

Reviewer #2: No

5. Review Comments to the Author

Reviewer #1: This is an interesting work from colleagues from Brazil providing insights on epidemiology and risk factors of hospitalised COVID-19 patients.

I have several comments for authors to consider

1. Can you provide percentage of people who were discarded due to unavailability of outcomes? This is important so as to mitigate selection bias.

2. Data analysis: How was the multiple imputation performed? Which software/platform were used?

3. Page 6: what do you mean by "with death. studied outcomes"??

4. Why was the predictive accuracy, sensitivity and ROC of the regression model not calculated?

Did you also account for the multicollinearity while selecting for variables to be included in the multivariate model? Please provide details?

5. The manuscript needs significant editing for language and comprehension.

For example, what do you mean by, "the unfamiliarity of the population with signs and symptoms of COVID-19 may have delayed healthcare seeking" (page 12) - healthcare seeking?

6. Several aspects of the results have not been discussed. Please organise the discussion so as to highlight key points and provide supporting discussion from relevant literature previously reported, if available.

7. In the discussion, suggest also discussing briefly that previous studies have reported prexisting factors such as diabetes are also associated with worse outcomes in COVID-19 patients ((see a metaanalysis by Bradley et al J Diabetes 2022 PMID: 34939735)

8. In the Discussion, you may also provide some indications, based on the findings and experiences from this study, some strategies which may be useful for future. Obviously role of telemedicine may also be discussed including streamlining telemedicine care for both hospitalised and non-hospitalised patients (see PMID: 33796502)

9. If possible, please also provide the ROC figure of the final prognostic model.

Reviewer #2: Dear Author(s),

Thank you for your esteemed efforts in increasing our collective knowledge about the predictors (risk factors) of mortality in hospitalized COVID-19 patients.

Please consider the following points:

1. Abstract: the "background" subsection is so broad and does not convey the objective of the study. Please re-write it.

2. Abstract: add explanation for "hospital LOS"

3. "A transmissible disease (COVID-19) caused by the SARS-CoV-2 virus increased the number of severe acute respiratory syndrome (SARS) cases worldwide since December 2019."

This sentence is not lingustically sound. COVID-19 is not an abbreviation for "a transmissible disease". SARS-CoV-2 is not explained. SARS refers to SARS-CoV-1, so please do not use it here.

4. Introduction: do not use expressions like "to date". Use actual dates to be specific.

5. "The United States of America and Brazil correspond to 71% of cumulative cases of COVID-19 in the Americas (more than 46 million cases)3." How this sentence is useful for your narrative?! US and Brazil are the most populous countries in the region, so it makes sense that they will have the highest number of cases/fatalities.

6. Why did you the age of 65 years old as cut-off point for analysis?

7. How did you decide whether the patient was critically or moderately ill on admission? Which categorization scheme have you used? Please mention that in your Methods section.

8. Age (p = 0.001), dyspnea (p = 0.003), and presence of other heart diseases (p =0.005) were different between groups."

It is unclear which groups do you refer to?

9. In Table 1; you did not add percentages of signs and symptoms. Please add them.

10. In Table 1: you did not add percentages of comorbidities. Please add them.

11. According to your data, the difference between females and males in mortality (32.3% vs. 30.7%) is neither statisitically or clinically signifcant. I do not understand why you insisted to say that females had higher mortality risk than males. It is not true based on your data. Please fix this accross your manuscript.

12. In your limitations section, you did not reflect on your information source (eCOVID-19 platform). Is it mandatory that all hospitalized patients must be registered in this platform? If the answer is "No", then this is an obvious limitation of your dataset that makes it prone to selection bias.

13. Your study design can not be called "retrospective cohort study", because hisotical/retrospective cohort studies require at least two distinct groups of individuals/patients to be involved in the study from the begining. You did something like chart review not a cohort study.

You can not consider the dichtomoy of survivor vs. non-survivor as a justification for calling your study as a cohort because the ground idea of cohort designs is dividing the groups of individuals based on their exposure/predictor/independent variable not their outcome/result/dependent variable.

For the outcome-based comparisons, we perform case-control studies not cohort studies.

14. Your Discussion might benefit from reflecting on hospitalization-related complications:

https://doi.org/10.3390/jcm10040581

Sincerely,

6. PLOS authors have the option to publish the peer review history of their article (what does this mean?). If published, this will include your full peer review and any attached files.

Reviewer #1: No

Reviewer #2: No

---

## [Author Response · Author response to Decision Letter 0]

1 Aug 2022

Thank you for your consideration and comments.

All adjustments were made to adapt to the journal's standards as requested by the editor in comments 1 to 4.

Below are responses to reviewers.

We would like to thank all considerations made by the reviewers. We clarified and modified the manuscript as suggested. We also appreciate the opportunity to send this new version for peer review.

Reviewer #1: This is an interesting work from colleagues from Brazil providing insights on epidemiology and risk factors of hospitalized COVID-19 patients.

I have several comments for authors to consider

1. Can you provide percentage of people who were discarded due to unavailability of outcomes? This is important so as to mitigate selection bias.

A total of 365 patients were registered on the eCOVID platform. Due to delays in seeking hospital care, lack of information, and unavailability of tests in Brazil, some patients may have undergone the exam outside the ideal window for virus identification. These patients were treated for COVID-19 but only based on the evolution of the clinical picture. We included only those records of patients who had a positive RT-PCR test for COVID-19 (N=280; 76.7%) since this was considered the gold standard for diagnosing COVID-19 worldwide at the time of data collection. 

2. Data analysis: How was the multiple imputation performed? Which software/platform were used?

Thank you for your question. All statistical analyses were conducted using the SPSS statistical software, version 22 (SPSS, Inc., Chicago, IL, USA). Missing data were managed using multiple imputations by replacing missing cells with random values from a statistical model based on distribution and underlying assumptions on the nature of the missing data. Five imputed datasets were created, and the final estimates were the average of different sets of imputed estimates and standard errors. The multiple imputation approach was chosen because it generates more accurate values than those generated by single imputation methods.

The above information has also been included in the revised text.

3. Page 6: what do you mean by "with death. studied outcomes"??

Thank you for the comment. This typo was corrected in the new version.

4. Why was the predictive accuracy, sensitivity and ROC of the regression model not calculated? Did you also account for the multicollinearity while selecting for variables to be included in the multivariate model? Please provide details?

Thank you for the suggestion. We have now provided the ROC curve of the regression model in Figure 2. The ROC curve showed a 0.84 predictive capacity for the computed regression model.

Regarding multicollinearity, this was not a problem in our analysis. We used the stepwise regression method to compute the model since it is used to provide an initial screening of variables within a large set of variables; thus, accounting for multicollinearity. We specified the predictors we would like to include, while the SPSS screened which factors contributed to predicting our dependent variable and excluded those who did not. Thus, we usually end up with fewer predictors than we specify. However, those that remain in the model tend to have solid and significant β-coefficients in the expected direction.

5. The manuscript needs significant editing for language and comprehension. For example, what do you mean by, "the unfamiliarity of the population with signs and symptoms of COVID-19 may have delayed healthcare seeking" (page 12) - healthcare seeking?

Thank you for your comment. The scientific writing of the manuscript was revised, and all inconsistencies were corrected.

The term “delayed healthcare seeking” was also corrected for “delayed health care seeking” (with a space) and maintained in the text because it is common in the literature (doi: 10.2147/HIV.S210977; doi: 10.1177/1049732315588083; doi: 10.4102/phcfm.v9i1.1378). 

6. Several aspects of the results have not been discussed. Please organise the discussion so as to highlight key points and provide supporting discussion from relevant literature previously reported, if available.

7. In the discussion, suggest also discussing briefly that previous studies have reported prexisting factors such as diabetes are also associated with worse outcomes in COVID-19 patients ((see a metaanalysis by Bradley et al J Diabetes 2022 PMID: 34939735)

8. In the Discussion, you may also provide some indications, based on the findings and experiences from this study, some strategies which may be useful for future. Obviously role of telemedicine may also be discussed including streamlining telemedicine care for both hospitalised and non-hospitalised patients (see PMID: 33796502)

Thank you for comments 6, 7, and 8. Below you can find the changes made to improve the understanding of the discussion. 

All key points of the results were included in paragraph 1: epidemiological aspects, comorbidities, signs and symptoms, and mortality. 

To optimize the discussion, new information was added in paragraph 4 (page 11). 

To answer comment 8, information was added in the last paragraph of the discussion. Telemedicine was not included as a strategy, given difficulties for implementation in Brazil and the low adherence of public health systems. Thus, this would not be a viable strategy from an economic point of view according to the current situation in the country. 

9. If possible, please also provide the ROC figure of the final prognostic model.

Thank you for the comment. The ROC figure of the final regression model has been provided (Figure 2).

Reviewer #2: Dear Author(s),

Thank you for your esteemed efforts in increasing our collective knowledge about the predictors (risk factors) of mortality in hospitalized COVID-19 patients.

Please consider the following points:

1. Abstract: the "background" subsection is so broad and does not convey the objective of the study. Please re-write it.

Thanks for the comment. Adjustments were made to the abstract.

2. Abstract: add explanation for "hospital LOS"

Thank you for the comment. Information about the term was added on page 5 (Variables and data collection).

3. "A transmissible disease (COVID-19) caused by the SARS-CoV-2 virus increased the number of severe acute respiratory syndrome (SARS) cases worldwide since December 2019."

This sentence is not linguistically sound. COVID-19 is not an abbreviation for "a transmissible disease". SARS-CoV-2 is not explained. SARS refers to SARS-CoV-1, so please do not use it here.

Thank you for the comment. We made the necessary adjustments to improve the understanding of the text.

4. Introduction: do not use expressions like "to date". Use actual dates to be specific.

Thank you! The text regarding this epidemiological data was updated.

5. "The United States of America and Brazil correspond to 71% of cumulative cases of COVID-19 in the Americas (more than 46 million cases)3." How this sentence is useful for your narrative?! US and Brazil are the most populous countries in the region, so it makes sense that they will have the highest number of cases/fatalities.

Thank you! The text regarding this epidemiological data was updated.

6. Why did you the age of 65 years old as cut-off point for analysis?

Thank you for your comment. It is well documented that the risk of severe COVID-19 increases with age. Since the CDC stated that adults aged ≥ 65 years were at greater risk of COVID-19 complications and death (https://www.cdc.gov/aging/covid19-guidance.html), we have decided to use this cut-off point to stratify individuals into younger vs. older adults. Other studies have also used this age as a cut-off for different analyses, as discussed in the manuscript.

7. How did you decide whether the patient was critically or moderately ill on admission? Which categorization scheme have you used? Please mention that in your Methods section.

Thank you for the comment. The information was added in the “Variables and data collection” of the methods section (page 5).

8. Age (p = 0.001), dyspnea (p = 0.003), and presence of other heart diseases (p =0.005) were different between groups."

It is unclear which groups do you refer to?

Thank you. We explain the subdivision of groups in the second paragraph of the Data Analysis section (Survivors and Non-survivors) and also in the revised paragraph. The aforementioned analyzes were conducted using these two groups, as shown in Table 1.

9. In Table 1; you did not add percentages of signs and symptoms. Please add them.

Thank you for the comment. All signs and symptoms presented in Table 1 (first column) were reported as percentage, as observed by the symbol (%) located next to the referred comorbidity.

10. In Table 1: you did not add percentages of comorbidities. Please add them.

Thank you for the comment. All comorbidities presented in Table 1 (first column) were reported as percentage, as observed by the symbol (%) located next to the referred comorbidity.

11. According to your data, the difference between females and males in mortality (32.3% vs. 30.7%) is neither statisitically or clinically signifcant. I do not understand why you insisted to say that females had higher mortality risk than males. It is not true based on your data. Please fix this accross your manuscript.

Thank you for the comment. Indeed, the data presented in the aforementioned comment indicate that mortality was higher in females than males; however, without statistical significance. In addition, paragraph 5 of the discussion presents the hypothesis that this higher death rate in females was probably due to particularities unrelated to sex. What we discussed in the manuscript concerns survival rate and survival analysis, in which we observed that hospitalized females had a lower survival rate than males.

12. In your limitations section, you did not reflect on your information source (eCOVID-19 platform). Is it mandatory that all hospitalized patients must be registered in this platform? If the answer is "No", then this is an obvious limitation of your dataset that makes it prone to selection bias.

Thank you for your comment. We included information about this limitation in the last paragraph of the discussion section.

13. Your study design can not be called "retrospective cohort study", because hisotical/retrospective cohort studies require at least two distinct groups of individuals/patients to be involved in the study from the begining. You did something like chart review not a cohort study.

You can not consider the dichtomoy of survivor vs. non-survivor as a justification for calling your study as a cohort because the ground idea of cohort designs is dividing the groups of individuals based on their exposure/predictor/independent variable not their outcome/result/dependent variable. For the outcome-based comparisons, we perform case-control studies not cohort studies.

Thank you for the comment. The modification was made in the appropriate place in the methods section (first paragraph, page 4).

14. Your Discussion might benefit from reflecting on hospitalization-related complications:

https://doi.org/10.3390/jcm10040581

Thank you for the comment. We adjusted the discussion section to improve the manuscript and the narrative of the results.

Sincerely,

---

## [Decision Letter · Decision Letter 1]

15 Aug 2022

PONE-D-22-12861R1Risk factors and survival in patients with COVID-19 in northeastern Brazil.PLOS ONE

Dear Dr. Fernandes,

Thank you for submitting your manuscript to PLOS ONE. After careful consideration, we feel that it has merit but does not fully meet PLOS ONE’s publication criteria as it currently stands. Therefore, we invite you to submit a revised version of the manuscript that addresses the points raised during the review process.

There are 2 major points about the study design raised by a reviewer, please pay attention to these.

We look forward to receiving your revised manuscript.

Kind regards,

Claudio Andaloro

Academic Editor

PLOS ONE

Reviewers' comments:

Reviewer's Responses to Questions

**Comments to the Author**

1. If the authors have adequately addressed your comments raised in a previous round of review and you feel that this manuscript is now acceptable for publication, you may indicate that here to bypass the “Comments to the Author” section, enter your conflict of interest statement in the “Confidential to Editor” section, and submit your "Accept" recommendation.

Reviewer #1: All comments have been addressed

Reviewer #2: (No Response)

2. Is the manuscript technically sound, and do the data support the conclusions?

Reviewer #1: Yes

Reviewer #2: Yes

3. Has the statistical analysis been performed appropriately and rigorously? 

Reviewer #1: Yes

Reviewer #2: Yes

4. Have the authors made all data underlying the findings in their manuscript fully available?

Reviewer #1: Yes

Reviewer #2: Yes

5. Is the manuscript presented in an intelligible fashion and written in standard English?

Reviewer #1: Yes

Reviewer #2: Yes

6. Review Comments to the Author

Reviewer #1: The revised manuscript has significantly improved. No further comments. I recommend the manuscript be accepted

Reviewer #2: Dear authors,

Thank you for your esteemed efforts in addressing my previous comments. Unfortunately, there is a couple of critical points that were not properly addressed.

1. Study design:

Case-control studies are retrospective in nature, there is no way to conduct a case-control study prospectively. The reason for this is very simple, distribution of subjects in the study arms "cases" and "controls" is made according to presence of the outcome of interest. The group of individuals who developed the outcome are described as cases, while those who did not experience the outcome are called controls. Therefore, you can not say "retrospective case-control study".

Also, I can't see any elements of case-control design in your workflow. For example, how and from where did you select the controls? which technique did you use for matching?

Your study is more or less a chart review = retrospective analysis for patients records.

2. If you would like to use the STROBE guidelines, please make sure to follow it strictly. I can see multiple discrepancies between your manuscript structure and the STROBE checklist.

Sincerely,

7. PLOS authors have the option to publish the peer review history of their article (what does this mean?). If published, this will include your full peer review and any attached files.

Reviewer #1: No

Reviewer #2: No

---

## [Author Response · Author response to Decision Letter 1]

29 Aug 2022

We would like to thank all considerations made by the reviewers. We clarified and modified the manuscript as suggested. We also appreciate the opportunity to send this new version for peer review.

Reviewer #2: Dear authors,

Thank you for your esteemed efforts in addressing my previous comments. Unfortunately, there is a couple of critical points that were not properly addressed.

1. Study design:

Case-control studies are retrospective in nature, there is no way to conduct a case-control study prospectively. The reason for this is very simple, distribution of subjects in the study arms "cases" and "controls" is made according to presence of the outcome of interest. The group of individuals who developed the outcome are described as cases, while those who did not experience the outcome are called controls. Therefore, you can not say "retrospective case-control study".

Also, I can't see any elements of case-control design in your workflow. For example, how and from where did you select the controls? which technique did you use for matching?

Your study is more or less a chart review = retrospective analysis for patients records.

Response: Thanks for the comment. We updated the study design section with the following: “A retrospective observational cross-sectional study was conducted.”(page 4)

2. If you would like to use the STROBE guidelines, please make sure to follow it strictly. I can see multiple discrepancies between your manuscript structure and the STROBE checklist.

Response: Thanks for the comment. Below is the STROBE check list (the "page" column indicates in which part of the manuscript the information can be found) used, some items were not performed by limitation or were not applicable. Because of this, it is more appropriate to remove from the text the part where the use of the STROBE checklist is mentioned (page 4)

Sincerely

---

## [Decision Letter · Decision Letter 2]

4 Oct 2022

PONE-D-22-12861R2Risk factors and survival in patients with COVID-19 in northeastern Brazil.PLOS ONE

Dear Dr. Fernandes,

Thank you for submitting your manuscript to PLOS ONE. After careful consideration, we feel that it has merit but does not fully meet PLOS ONE’s publication criteria as it currently stands. Therefore, we invite you to submit a revised version of the manuscript that addresses the points raised during the review process.

Authors should determine the design of their study properly, as pointed by reviewer 2. Please pay attention to the doubts raised.

We look forward to receiving your revised manuscript.

Kind regards,

Claudio Andaloro

Academic Editor

PLOS ONE

Journal Requirements:

Reviewers' comments:

Reviewer's Responses to Questions

**Comments to the Author**

1. If the authors have adequately addressed your comments raised in a previous round of review and you feel that this manuscript is now acceptable for publication, you may indicate that here to bypass the “Comments to the Author” section, enter your conflict of interest statement in the “Confidential to Editor” section, and submit your "Accept" recommendation.

Reviewer #1: All comments have been addressed

Reviewer #2: All comments have been addressed

2. Is the manuscript technically sound, and do the data support the conclusions?

Reviewer #1: Yes

Reviewer #2: Yes

3. Has the statistical analysis been performed appropriately and rigorously? 

Reviewer #1: Yes

Reviewer #2: Yes

4. Have the authors made all data underlying the findings in their manuscript fully available?

Reviewer #1: Yes

Reviewer #2: Yes

5. Is the manuscript presented in an intelligible fashion and written in standard English?

Reviewer #1: Yes

Reviewer #2: Yes

6. Review Comments to the Author

Reviewer #1: The authors have adequately addressed the comments. I recommend the manuscript be accepted in its current form.

Reviewer #2: 1. Cross-sectional studies can not be described as prospective or retrospective. This is a common mistake.

2. As you said your study is cross-sectional, you do not need to say "observational".

3. The two main types of cross-sectional studies are "descriptive vs. analytical":

https://obgyn.onlinelibrary.wiley.com/doi/full/10.1111/aogs.13331

4. Please update the study design in your abstract accordingly.

5. The study time frame is missed.

6. I still believe that your study is better described as a "retrospective cohort" study.

7. PLOS authors have the option to publish the peer review history of their article (what does this mean?). If published, this will include your full peer review and any attached files.

Reviewer #1: **Yes: **Sonu Bhaskar

Reviewer #2: No

---

## [Author Response · Author response to Decision Letter 2]

20 Oct 2022

Journal Requirements:

Response: Thanks for the comments. Below are the changes related to the list of references.

1) A correction was found in relation to reference number 8 (Zhou F, Yu T, Du R, Fan G, Liu Y, Liu Z, et al. Clinical course and risk factors for mortality of adult inpatients with COVID-19 in Wuhan, China: a retrospective cohort study. Lancet. 2020;395(10229):1054-62. doi: 10.1016/s0140-6736(20)30566-3. PMCPMC7270627). In this Article, the units for d-dimer, haemoglobin, and high-sensitivity cardiac troponin I have been corrected to μg/mL (d-dimer), g/L (haemoglobin), and pg/mL (high-sensitivity cardiac troponin I). In figure 1, the start of systematic corticosteroid for non-survivors has been changed to day 13 after illness onset. The appendix has also been corrected. These corrections have been made to the online version as of March 12, 2020, and will be made to the printed version

2) The Original Investigation titled “Baseline Characteristics and Outcomes of 1591 Patients Infected With SARS-CoV-2 Admitted to ICUs of the Lombardy Region, Italy,” (number 13 on the reference list) published April 6, 2020, has been corrected to include the nonauthor collaborator (group) names in a supplement. The doi number remains the same as the first version.

3) In relation to reference 17 titled “Salinas-Escudero G, Carrillo-Vega MF, Granados-Garcia V, Martinez-Valverde S, Toledano-Toledano F, Garduno-Espinosa J. A survival analysis of COVID-19 in the Mexican population. BMC Public Health. 2020;20(1):1616. doi: 10.1186/s12889-020-09721-2.” An amendment to this paper has been published and can be accessed via the original article. Because of this, was included the document (number 18 on reference list) where the correction can be found.

 Reviewer #2: 

1. Cross-sectional studies can not be described as prospective or retrospective. This is a common mistake.

2. As you said your study is cross-sectional, you do not need to say "observational".

3. The two main types of cross-sectional studies are "descriptive vs. analytical":

https://obgyn.onlinelibrary.wiley.com/doi/full/10.1111/aogs.13331

4. Please update the study design in your abstract accordingly.

5. The study time frame is missed.

6. I still believe that your study is better described as a "retrospective cohort" study

Response: Thanks for the comments. After analyzing the suggested references (comment 3) and observing what is postulated in the book " Foundation of Clinical Research: Applications to practice. Portney, L.G & Watkins, M.P. Third edition. F.A Davis Company, 2015" , this research ranks more appropriately in a retrospective cohort. The necessary modification was made on page 4 and in the abstract (page 2).

The time frame of the study was included on page 4 (lines 2 and 3). 

Sincerely,

---

## [Decision Letter · Decision Letter 3]

14 Nov 2022

Risk factors and survival in patients with COVID-19 in northeastern Brazil.

PONE-D-22-12861R3

Dear Dr. Fernandes,

We’re pleased to inform you that your manuscript has been judged scientifically suitable for publication and will be formally accepted for publication once it meets all outstanding technical requirements.

Kind regards,

Claudio Andaloro

Academic Editor

PLOS ONE

Additional Editor Comments (optional):

Reviewers' comments:

Reviewer's Responses to Questions

**Comments to the Author**

1. If the authors have adequately addressed your comments raised in a previous round of review and you feel that this manuscript is now acceptable for publication, you may indicate that here to bypass the “Comments to the Author” section, enter your conflict of interest statement in the “Confidential to Editor” section, and submit your "Accept" recommendation.

Reviewer #1: All comments have been addressed

Reviewer #2: All comments have been addressed

2. Is the manuscript technically sound, and do the data support the conclusions?

Reviewer #1: Yes

Reviewer #2: Yes

3. Has the statistical analysis been performed appropriately and rigorously? 

Reviewer #1: Yes

Reviewer #2: Yes

4. Have the authors made all data underlying the findings in their manuscript fully available?

Reviewer #1: Yes

Reviewer #2: No

5. Is the manuscript presented in an intelligible fashion and written in standard English?

Reviewer #1: Yes

Reviewer #2: Yes

6. Review Comments to the Author

Reviewer #1: No further comments. The authors have addressed the comments. I recommend the manuscript be accepted.

Reviewer #2: Dear author(s),

Thank you for addressing all my previous comments. I believe that the manuscript is in a good shape now.

Sincerely,

7. PLOS authors have the option to publish the peer review history of their article (what does this mean?). If published, this will include your full peer review and any attached files.

Reviewer #1: No

Reviewer #2: No

---

## [Editor Report · Acceptance letter]

16 Nov 2022

PONE-D-22-12861R3 

Risk factors and survival in patients with COVID-19 in northeastern Brazil. 

Dear Dr. Fernandes:

I'm pleased to inform you that your manuscript has been deemed suitable for publication in PLOS ONE. Congratulations! Your manuscript is now with our production department. 

Kind regards, 

on behalf of

Dr. Claudio Andaloro 

Academic Editor

PLOS ONE